# Treatment of *Ginkgo biloba* with Exogenous Sodium Selenite Affects Its Physiological Growth, Changes Its Phytohormones, and Synthesizes Its Terpene Lactones

**DOI:** 10.3390/molecules27217548

**Published:** 2022-11-03

**Authors:** Linling Li, Jie Yu, Li Li, Shen Rao, Shuai Wu, Shiyan Wang, Shuiyuan Cheng, Hua Cheng

**Affiliations:** 1School of Modern Industry for Selenium Science and Engineering, Wuhan Polytechnic University, Wuhan 430023, China; 2National R&D Center for Se-Rich Agricultural Products Processing, Wuhan Polytechnic University, Wuhan 430023, China

**Keywords:** ginkgolides, metabolome, physiological index, sodium selenite, transcriptome

## Abstract

Ginkgolide is a unique terpenoid natural compound in *Ginkgo biloba*, and it has an important medicinal value. Proper selenium has been reported to promote plant growth and development, and improve plant quality, stress resistance, and disease resistance. In order to study the effects of exogenous selenium (Se) on the physiological growth and the content of terpene triolactones (TTLs) in *G. biloba* seedlings, the seedlings in this work were treated with Na_2_SeO_3_. Then, the physiological indexes, the content of the TTLs, and the expression of the related genes were determined. The results showed that a low dose of Na_2_SeO_3_ was beneficial to plant photosynthesis as it promoted the growth of ginkgo seedlings and increased the root to shoot ratio. Foliar Se application significantly increased the content of soluble sugar and protein and promoted the content of TTLs in ginkgo leaves; indeed, it reached the maximum value of 7.95 mg/g in the ninth week, whereas the application of Se to the roots inhibited the synthesis of TTLs. Transcriptome analysis showed that foliar Se application promoted the expression levels of *GbMECPs*, *GbMECT*, *GbHMGR*, and *GbMVD* genes, whereas its application to the roots promoted the expression of *GbDXS* and *GbDXR* genes. The combined analysis results of metabolome and transcriptome showed that genes such as *GbDXS*, *GbDXR*, *GbHMGR*, *GbMECPs*, and *GbCYP450* were significantly positively correlated with transcription factors (TFs) *GbWRKY* and *GbAP2*/*ERF*, and they were also positively correlated with the contents of terpene lactones (ginkgolide A, ginkgolide B, ginkgolide M, and bilobalide). Endogenous hormones (MeJA-ILE, ETH, and GA7) were also involved in this process. The results suggested that Na_2_SeO_3_ treatment affected the transcription factors related to the regulation of endogenous hormones in *G. biloba*, and further regulated the expression of genes related to the terpene synthesis structure, thus promoting the synthesis of ginkgo TTLs.

## 1. Introduction

*G. biloba* is the only surviving representative of Ginkgoaceae, which originated approximately 280 million years ago. It is an important economic tree species with economic, ornamental, and medicinal values [1]. *G. biloba* extract can relieve asthma, activate blood circulation, remove blood stasis, and relieve pain. It is used to treat coronary heart disease, angina pectoris, hyperlipidemia, and so on. Various drugs manufactured from *G. biloba* leaves and seeds have been widely found in pharmacies and hospitals (e.g., dipyridamole injections), whose main bioactive components are ginkgolides and flavonoids, have been widely used in the clinical treatment of cardiovascular and cerebrovascular diseases [2]. TTLs mainly include ginkgolides A, B, C, M, and bilobalide. Ginkgolides have the physiological functions of antioxidation, they antagonize active platelet factors, they protect the nervous system, they are anti-inflammatory, they prevent tumors, and so on [3]; however, the TTL content of *G. biloba* is quite low, and it fails to meet the demands of pharmaceutical production [4]. Ginkgolides are terpenoids, and there are two main pathways involved in their precursor biosynthesis, including the mevalonate (MVA) pathway in the cytoplasm and the 2-C-methyl-d-erythritol-4-phosphate (MEP) pathway in the plastids [5]. At present, it is believed that the biosynthesis of ginkgolide precursors mainly occurs through the MVA pathway, and the biosynthesis of the ginkgo diterpene lactone precursors mainly occur through the MEP pathway (Figure 1) [6]. In the MEP pathway, *DXS*, *DXR*, *HDR*, *LPS*, *CYP450,* and other genes are considered to be the key enzyme genes regulating ginkgolide synthesis [7,8,9]. Most genes involved with the ginkgolide biosynthesis pathway have been cloned, among which, the more important genes are *GbHMGR*, *GbDXS*, *GbDXR*, *GbHDR*, *GbGGPPS*, *GbFPS,* and *GbLPS*. *GbHMGR* belongs to a small gene family, and its gene expression is tissue specific. It is only expressed at a low level in the root and it is induced by phytohormones such as JA [10,11]. *GbDXS* also plays an important role in the biosynthesis of ginkgolides, and it is only expressed in the radicle [8]. Tissue expression analysis showed that the expression level of *GbDXR* in roots was higher than that in leaves, which further proved that ginkgolides were biosynthetic in roots [7,8].

Inducible factors can stimulate the expression of key genes in the plant biosynthesis pathway, thus regulating the synthesis of secondary metabolites [12]. Depending on the source, inducers can be divided into biological inducers and abiotic inducers. Biological inducers mainly include plant cell components and microorganisms, whereas abiotic inducers refer to metal ions and inorganic substances that can trigger plant defense mechanisms. Abiotic elicitors have been widely used to regulate the biosynthesis of secondary metabolites in medicinal plants [5,13]. At present, the abiotic elicitors mainly include methyl jasmonate, salicylic acid, heavy metal salts, and rare-earth elements. Selenium (Se) is a trace nutrient that is essential for living organisms, due to its fundamental involvement in several physiological and metabolic processes [14]. Se also has an important role in plants, as it promotes growth and development at suitable concentrations [15]. Se treatments have been reported to lead to a significant improvement in vegetative growth and photosynthetic pigment accumulation in plants. Furthermore, some investigations have indicated that Se at appropriate levels can partially reduce chloroplast degradation and increase chlorophyll content [16]. In addition, selenium can affect the synthesis of plant secondary metabolites, such as flavonoids, phenols, and other substances, thus improving the nutritional value and quality of crops [17,18,19]. At present, there are few reports on the effects of Se on plant terpenoids.

In this study, *G. biloba* seedlings were treated with Na_2_SeO_3_ at different concentrations. The growth parameters, physiological indicators, TLLs content, and expression of related genes were determined in order to reveal the possible regulatory relationship between hormones, TFs, and TLL synthesis-related genes. The research results were expected to provide theoretical data for the better use of selenium biofortification in order to enhance the total terpenoid content in ginkgo leaves.

## 2. Results

### 2.1. Effects of Exogenous Na_2_SeO_3_ on the Physiological Indexes of Ginkgo biloba

Na_2_SeO_3_ treatments affected the biomass of *G. biloba* leaves. Spraying a certain concentration of Na_2_SeO_3_ on a liquid surface was beneficial to the yield of ginkgo leaves, whereas the application of Se to the roots did not significantly improve the yield of leaves (Figure 2a). The change in Se concentration on the leaf surface first led to an increase in biomass followed by a decrease. When the concentration of Se applied on the leaves was 15 mg/L, the biomass was about 1.76 times that of the control. As the concentration increased, burning spots appeared at a concentration of 25 mg/L, and the biomass decreased, becoming lower than that of the control group. Its application to the roots showed that it had a certain effect on the seedling biomass that encouraged its growth, and it was 1.31 times that of the control when the concentration reached 20 mg/L; however the biomass began to decrease as the concentration continued to increase.

The different ways of applying Se have a great influence on the root/shoot ratio of ginkgo seedlings (Figure 2b). Regarding the application of Se to the leaves, the concentration of Se was first decreased and then increased; however, when Se was applied to the roots, the concentration of Se was first increased and then decreased. When the concentration of Se was 30 mg/L, when it was applied to the leaves, the root to shoot ratio was the greatest, at 1.12 times that of the control; when the concentration was 10 mg/L, it was the lowest, at 0.59 times that of the control. When the concentration of Se was 15 mg/L, when it was applied to the roots, the root to shoot ratio was the greatest, at 1.1 times that of the control. With the increase in the concentration of the Se treatment, the root to shoot ratio gradually decreased, and when the concentration was 30 mg/L, the root to shoot ratio was the lowest, at 0.70 times that of the control.

The application of Na_2_SeO_3_ on the leaves significantly promoted the chlorophyll content of ginkgo leaves (Figure 2c). Compared with the control, different Se concentrations were shown to increase the leaves’ chlorophyll content. When the exogenous Na_2_SeO_3_ reached 25 mg/L, the chlorophyll content reached a maximum value of 1.15 mg/g, though it slightly decreased with further increases in concentration, and burning spots also appeared as the concentration continued to increase; however, when the concentration of Se applied to the roots was 15 mg/L, the chlorophyll content reached a maximum level of 0.72 mg/g, which then began to decline as the selenium concentration increased.

Na_2_SeO_3_ greatly influenced the soluble sugar content of leaves (Figure 2d). In the control group, the content of soluble sugar in *G. biloba* leaves gradually increased within 7 weeks, slightly decreased at 9 weeks, and then reverted back to the level that it was at 7 weeks. The soluble sugar content in the group where Se was applied to the leaves increased significantly in the first week after Se was applied; it then increased again in the seventh week. In the group where Se was applied to the roots, the soluble sugar content increased significantly in the third week, and then decreased gradually. In general, Se could promote the increase of soluble sugar content in leaves, though the increase was more obvious after the leaf treatment.

The soluble protein content in *G. biloba* leaves was significantly increased by foliar Se application (Figure 2e). During the third week of applying Se to the leaf surface, the soluble protein content was 2.61 times that of the control group. In the ninth week, it reached 3.82 times that of the control group, and the content of soluble protein reached its maximum during the eleventh week, at 2.25 times that of the control group. Nonetheless, the group where Se was applied to the roots showed a slight decrease in the early stages of treatment, followed by an increase in the later stages.

The foliar application of Se promoted the content of terpene lactones in ginkgo leaves, whereas the application of Se to the roots had an inhibitory effect (Figure 2f). The application of Se to the leaves showed an overall increase over time, and its content was 1.57 times higher than that of the control group in the fifth week, reaching a maximum value of 7.95 mg/g in the ninth week; however, after applying Se to the roots, the content of terpene lactones in *G. biloba* leaves decreased. In the ninth week, the content of terpene lactones in *G. biloba* leaves was only 0.73 times that in the control group.

### 2.2. Transcriptome Analysis of Ginkgo biloba Treated with Na_2_SeO_3_

Illumina HiSeq sequencing was carried out on ginkgo leaves treated with Na_2_SeO_3_, generating 447,207,439 raw reads, then 431,640,524 clean reads, after filtering out low-quality fragments (Appendix A). The average GC content of each sample was 47.64%, and the proportion of the base quality that exceeded Q30 was more than 90%, which indicated that the transcriptome sequencing quality and data volume were relatively high, thus meeting the requirements of subsequent analysis. The data statistics of each sample are shown in Appendix A. Since there was no reference genome, after obtaining clean reads, Trinity was used for splicing and assemblage, and the longest transcript of each gene was taken as unigene for subsequent analysis. A total of 142,834 unigenes were obtained after splicing and assembly, with an average length of 656 bp.

The results of the KOG classification (Figure 3) showed that 18,671 unigenes could be annotated with functional information, and 26 functional classifications could be annotated. Posttranslational modification, protein folding, and molecular chaperones (14.97%) were the most widely covered, followed by translation, ribosome structure, and biosynthesis (12.4%), and secondary metabolite synthesis, transport, and metabolism (10.7%). In all categories, nucleic acid structure accounted for (0.06%), cell movement accounted for (0.03%), and unknown proteins accounted for (0.006%).

A total of 5292 DEGs were found by differential expression screening analysis of unigenes from three groups of samples, which accounted for 3.7% of the total annotated unigene. When the unigene was compared with the KEGG database, 25,479 genes were annotated, and 130 pathways were involved. These annotations included statistics concerning the first 20 pathways of differential gene enrichment in the CK group, group A, and group B. These groups were mainly enriched with phenylpropanoid biosynthesis, plant hormone signal transduction, amino acid and nucleotide sugar metabolisms, and starch and sucrose metabolisms. The annotation analysis is shown in Figure 4. Moreover, the pathways related to the synthesis of terpenoids mainly included terpenoid skeleton biosynthesis, sesquiterpene and triterpene biosynthesis, monoterpenoid biosynthesis, diterpene biosynthesis, ubiquinone and other terpenoid quinone biosynthesis. Using the DESeq algorithm to analyze the differences between unigenes in three groups of samples, the screening threshold was padj < 0.05. The results showed that compared with the control group, there were 4653 differential genes in group A, including 2934 upregulated genes and 1719 downregulated genes, whereas group B had 2546 differential genes, including 1920 upregulated genes and 626 downregulated genes. There were only 394 differential genes between Group A and Group B, including 244 upregulated and 150 downregulated genes (Figure 5).

### 2.3. Differential Analysis of Metabolic Groups

Through the detection of metabolites in the three groups of samples, 991 metabolites were detected, including known and unknown metabolites. After classification, 29 kinds of metabolites were obtained, including flavonoids, sugars, amino acids, terpenoids, among others. Moreover, 15 kinds of terpenoids were detected, including ginkgolide A, ginkgolide B, ginkgolide C, ginkgolide J, ginkgolide M, gentiopicroside, zeaxanthin, plant cypermethrin C, zeaxanthin, dioscin, caffeinol juniperidine, rice diterpene phytoalexin, and plant cypress D.

By screening the differential metabolites of the three groups of samples, there were 315 differential metabolites between the control group and treatment group A, 324 differential metabolites between the control group and treatment group B, and only 154 differential metabolites between treatment group A and treatment group B (Figure 6). Statistics concerning the number of different metabolites in each group are shown in Appendix A.

The KEGG database was used to annotate and display the significantly different metabolites, and the annotation results were classified according to the types of pathways in KEGG, as shown in Figure 7. There were 215 significantly different metabolites in treatment group A and the control group, and they enriched 79 pathways, mainly those concentrated in the metabolic pathway, flavonoids and flavonol biosynthesis, the biosynthesis of secondary metabolites, and the ABC transport pathway. Moreover, 247 significantly different metabolites in treatment group B and the control group enriched 83 pathways that were mainly concentrated in the metabolic pathway, the biosynthesis of secondary metabolites, flavonoids and flavonol biosynthesis, and the ABC transporter pathway. The most significant difference between treatment group B and treatment group A is the fact that 107 metabolites enriched 48 pathways that were mainly concentrated in the biosynthesis of secondary metabolites and metabolic pathways. Only diterpenoid biosynthesis was involved in the metabolic pathway of terpenoid biosynthesis, and only gibberellin A3 (GA3) and GA7 were involved in the metabolic pathway of terpenoid biosynthesis.

### 2.4. Kmeans Analysis of Transcriptome and Metabolome

The transcriptome and metabolome data of the three sample groups were analyzed using Kmeans clustering. A total of 30 cluster groups were obtained for differentially expressed genes and 10 were obtained for differentially expressed metabolites (Appendix A). Among them, the eighth type of DEG, and the tenth type of metabolite, had similar patterns. The 21st type of differentially expressed gene was similar to the 7th type of metabolite (Appendix A). Considering that the terpenoid metabolites of *G. biloba* were mainly concentrated in group 6, the corresponding differentially expressed genes were mainly concentrated in group 14. The 14th group of differentially expressed genes contained 76 differentially expressed genes in total, and the sixth group of differentially expressed metabolites contained six differentially expressed metabolites. We selected the 14th class of differentially expressed genes and the 6th class of differentially expressed metabolites for further analysis, and the relationship between differentially expressed genes and differentially expressed metabolites was examined.

### 2.5. Association Network Analysis of DEGs and Terpene Lactones

In order to better explore the relationship between ginkgolide synthesis, hormones, and related metabolic genes, the transcriptome and metabolome of different ginkgo leaves were conjointly analyzed (Figure 8).

The screened metabolites were associated with key terpene synthesis genes and transcription factors. The results showed that ginkgolide A (GbL0824) had a significant positive correlation with the *GbDXR* gene (TRINITY_DN50519_c0_g5, ρ = 0.9017, *p*-value = 0.00035) and *GbHMGR* (TRINITY_DN51177_c0_g5, ρ = 0.9333, *p*-value = 0.00074). There was a significant positive correlation with the transcription factor *AP2*/*ERF* (ρ = 0.9246, *p*-value = 0.00089) and with phytohormone MeJA-ILE (ρ = 0.8281, *p*-value = 0.0058).

Ginkgolide B (GbL0799) content was positively correlated with *GbDXR* (ρ = 0.9136, *p*-value = 0.00057) and the *GbDXS* (TRINITY_DN49098_c0_g2, ρ = 0.8667, *p*-value = 0.00450) gene expression level, as well as with the transcription factor *WRKY57* (TRINITY_DN51893_c1ug3, ρ = 0.8897, *p*-value = 0.0013), and phytohormone ethylene (ETH, ρ = 0.7574, *p*-value = 0.0181).

Ginkgolide M (GbL0020) was positively correlated with CYP450724B1 (TRINITY_DN45314_c0_g1, ρ = 0.9249, *p*-value = 0.00035) and *GbMECPs* (TRINITY_DN49886_c1_g5, ρ = 0.8167, *p*-value = 0.0108), gene expression, as well as the transcription factor *WRKY41* (TRINITY_DN52998_c0_g9, ρ = 0.9358, *p*-value = 0.00021), and phytohormone GA7 (ρ = 0.9119, *p*-value = 0.0061).

The only sesquiterpene lactone, bilobalide (GbL0504), and the *CYP450_725* (TRINI-TY_DN52038_c0_g1, ρ = 0.9433, *p*-value = 0.00013) gene were significantly positively correlated. There was also a significant positive correlation with the transcription factor *WRKY41* (TRINITY_DN52998_c0_g9, ρ = 0.9252, *p*-value = 0.00035), and with phytohormone ETH (ρ = 0.8368, *p*-value = 0.0049).

### 2.6. Gene Expression Analysis of Key Enzymes in Terpenoid Synthesis

In the experiment, six different genes (Appendix A) that are related to terpenoid synthesis, and which are significantly different in the KEGG metabolic pathway, were selected for qRT-PCR verification, and the expression of these six genes in ginkgo leaves was analyzed (Figure 9).

*DXS* gene expression results showed that the expression of the *DXS* gene increased to the maximum level in the first week after Se was applied to the leaf surface, reaching a level that was approximately three times that of the control group. Regarding the following sampling points, in the third week, the *DXS* gene expression of the group where Se was applied to the leaves was significantly lower than that of the control group, but during the sampling point in the fifth week, the *DXS* gene expression was slightly higher than that of the control group, and there was no significant difference between the sampling points and the control group in the following weeks. In the group where Se was applied to the roots, *DXS* gene expression was significantly increased compared with the control group; in the fifth week, it reached levels that were approximately 14 times that of the control group. Aside from the upregulated expression that was observed in weeks 5 and 11, gene expression during other sampling points was lower compared with the control group.

In the first week after root treatment, the *DXR* expression level of the sampling point was the highest, and it was approximately seven times higher than that of the control group. In the fifth week, it was nearly five times higher than the control group, and other sampling points were the same as the control group. After root treatment, the *DXR* gene expression level was slightly increased in the fifth week compared with the control group, and the relative expression during other sampling points was significantly lower than in the control group.

In the first week after spraying Na_2_SeO_3_ on the leaf surface, the expression level of the *MECT* gene was slightly increased (2.7 times) compared with the control group. The maximum increase was observed in the third week, reaching levels about 28 times that of the control group. Then, the expression level was significantly reduced in the control group in the fifth week, it was reduced by 0.3 times in the seventh week, after which, an increase was observed in the control group. After root treatment, its expression level was lower than that of the control group in the first three weeks, but it increased by about four times in the fifth week, then, it decreased to 0.04 times in the seventh week, and then increased to the same level as the control group. In the eleventh week, the increase was the greatest, at approximately 5.5 times of that of the control group.

After the ginkgo leaves were treated with Na_2_SeO_3_, the expression level of MECP genes decreased slightly in the first three weeks compared with the control group, it rapidly increased to about 12 times that of the control group in the fifth week, it decreased to the same level as the control group in the seventh week, and it increased to a level that was three times that of the control group between the ninth and eleventh week. The expression level increased slightly in the first week after root treatment, and decreased below the level of the control group in the third week. It rose to 5.9 times that of the control group in the fifth week, after which, it remained at the same level as the control group. It increased to about four times that of the level of the control group in the eleventh week.

After the leaves were treated with Na_2_SeO_3_, the expression level of the *HMGR* gene slightly decreased, then increased to 3.3 times of the control group in the third week, and it reached the maximum in the fifth week, amounting to about 54 times that of the control group. Then, the expression level rapidly decreased, before increasing again in the ninth and eleventh weeks. After root treatment, the expression level decreased to below the level of the control group in the first three weeks, and then it increased to about 3.3 times the level of the control group in the fifth week. It then decreased, before increasing to 7.9 times the level of the control group in the ninth week, when it reached its peak. In the 11th week, the expression level was lower than that of the control group.

## 3. Discussion

The TLLs of ginkgo are highly effective platelet activating factor antagonists, which can highly selectively antagonize platelet aggregation and prevent thrombosis. They are widely used in the treatment of cardiovascular and cerebrovascular diseases [2,3]. Due to the difficulty and high cost of chemical synthesis technology, ginkgolides are mainly extracted from ginkgo leaves. Se is not a necessary trace element for plant growth, but it is a beneficial element for the cultivation and growth of many crops, which thus affects the biomass and growth quality of crops [20,21,22,23,24]. Furthermore, Se at both 3 and 5 µM concentrations triggered a 25% spike in terms of leaf area, which resulted in an increase in the plant’s overall growth and biomass [25]. In this study, different doses of Se were shown to affect the total biomass of *G. biloba* leaves. Low doses of Se can increase the biomass of leaves, and high doses of Se can reduce the biomass of leaves. The root application of Se promoted the increase of leaf biomass only when the concentration was above 20 mg/L (Figure 2a). This prompts speculation concerning the idea that an effective concentration of Se, directly applied to leaves, affects the aboveground parts of the plant, whereas an effective concentration of Se applied to the roots does not affect the aboveground parts of the plant, due to the dilution of soil and water. Studies have shown that, compared with the control group, 5 µg of Se promoted the root growth of pepper plants, and the relative water content increased by 13% [26]; therefore, the treatment of ginkgo roots with a low concentration of Se (≤15 mg/L) promoted the elongation of ginkgo roots, although the aboveground biomass did not increase, meaning that the root to shoot ratio increased (Figure 2b).

Plant photosynthetic physiology is particularly sensitive to environmental stress. It has been reported that Se can regulate environmental stress and further affect plant photosynthesis. In their study, Freeman et al. compared the electron transfer rate (ETR) of high Se plants and non-high Se plants under Se treatment, finding that high Se plants that were subjected to a 20 μM selenate treatment had significantly increased ETR, whereas non-Se plants had significantly decreased ETR under the same conditions [27]. It has been found that Se can regulate environmental stress and may further affect plant photosynthesis. Zhang et al. found that the application of Se (<50 g/hm^2^) to rice increased the photosynthetic rate (Pn), intercellular CO_2_ concentration (Ci), ETR photosynthetic index, chlorophyll fluorescence, F_v_, F_o_, and the F_v_/F_o_ of plants. When the amount of applied Se was 100 g/hm^2^, the photosynthetic index decreased [28]. The effect of the Se mechanism on plant photosynthesis may be similar to that of the antioxidant system. ROS accumulation is induced when the plant’s photosynthesis electron transmission is blocked. Se affects plant photosynthesis by directly or indirectly inhibiting or inducing the accumulation of ROS in plants and photosynthesis-related enzyme systems [29]. On the other hand, Se may affect the process of electron transfer and photosynthetic energy conversion by affecting Fe-S protein synthesis [30]. Recent studies have shown that 50–100.0 mg/L of exogenous Se can promote photosynthesis and the mineral element absorption of tea plants, and it can also improve their biomass. It can also promote the accumulation of polyphenols, theanine, flavonoids, and volatile secondary metabolites in tea, thereby improving the nutritional quality of tea [31]. In this study, a low-dose Se treatment increased the chlorophyll content of ginkgo leaves, promoted photosynthesis and the content of soluble sugar and soluble protein, and finally provided sufficient carbohydrates for an increase in the total terpene lactone content of leaves (Figure 2c,d). The increase in biomass is also an important factor for promoting the increase of total terpene lactone content in ginkgo leaves (Figure 2a,f).

At present, there are many reports concerning the effect of Se on a plant’s secondary metabolism, but most of them focus on the effect that Se has on the synthesis of flavonoids. In their study, Li et al. reported that the application of nano Se promoted the biosynthesis of flavonoids in celery, such as apigenin (58.4%), rutin (66.2%), ρ-coumarinic acid (80.4%), ferulic acid (68.2%), luteolin (87.0%), and kaempferol (105.7%) [19]. In *G. biloba*, the appropriate concentration of Se could promote the expression level of the flavonoid synthesis gene, further promoting the increase of total flavonoid content in *G. biloba* leaves, which is conducive to the medicinal value of ginkgo extract [18,32]. In a study focusing on the treatment of *Pueraria lobata* with Se, Se not only induced the expression of sulfate and phosphate transporters that are related to the metabolism of Se, but also the expression of 16 structural genes related to puerarin synthesis and a large number of miRNAs [33,34]. At present, there are few reports on the effects of Se on plant terpenoids. The field survey conducted in the natural Se agricultural area of Colorado in the United States found that the floral concentrations of medicinal cannabidiol (CBD) and terpenoids were not affected by Se in *Cannabis sativa* L. [35]; however, in the study of *G. biloba*, it was found that a low concentration of foliar Se promoted the increase of terpene lactone content in ginkgo leaves (Figure 2f).

The cytochrome P450s belong to CYP families, which diversify in pre-seed plants and gymnosperms, but are not preserved in angiosperms [36]. The results showed that CYP450 had an important role in the oxidation, cyclization, and rearrangement of multi-step products, downstream of ginkgolides. The CYP450 enzyme is the first step in the oxidation reaction of levopimaradiene and ginkgolides [9]. In recent years, regarding *WRKY*, *bZIP*, *bHLH,* and the *AP2*/*ERF* family, TFs have been continuously cloned, and they have been proven to have a key regulatory role in the biosynthesis of terpenoids [4,37,38]. DXR catalyzes DXP to generate MEP, which is an important rate limiting enzyme in the MEP pathway and an ideal target for regulating the MEP pathway [8]. In a study focusing on the *DXR* gene promoter of *G. biloba*, we found that WRKY, MYB, MYC, and AP2/ERF binding sites exist in its upstream regulatory elements, and AP2/ERF are important TFs regulating the expression of the *DXR* gene [39]. WRKYs were found to be involved in the regulation of other terpenoid synthesis structure genes in *G. biloba*.

The change in hormone level in ginkgo leaves will affect the content of terpene lactones in leaves. Exogenous GA treatments of *G. biloba* leaves increase the content of terpene lactones in leaves, and they also prolong the harvest time of leaves [40]. Results of evaporative light-scattering detector–high-performance liquid chromatography showed that ABA, SA, MeJA, and ETH treatments could increase the TTL content at various levels, and up to 21.9% in ginkgo leaves [13]. In the process of MeJA regulation, the JAZ signal transduction protein participates in the regulation of the TTL synthesis structure gene of *G. biloba* [41]. Interestingly, the expression of the ginkgo leaves’ *WRKYs* is regulated by many phytohormones, such as ABA, SA, MeJA, GA, and ETH. These hormones regulate ginkgo terpene lactone synthesis by affecting WRKYs [37,42]. A KEGG database comparison revealed that differentially expressed genes of different Se treatments were mainly concentrated in phenylpropanoid biosynthesis, phytohormone signal transduction, terpene synthesis and other related pathways (Figure 4). Furthermore, *K*-means analysis of transcriptome data and differential metabolites revealed that key genes, transcription factors, hormones, and so on, were involved in ginkgolide synthesis (Appendix A). The correlation network analysis of differentially expressed genes and metabolites showed that the synthesis of ginkgolides was significantly positively correlated with the contents of the endogenous hormones MeJA, GA7, and ETH in the leaves and roots of *G. biloba* that were treated with Na_2_SeO_3_ (Figure 8). It is possible that an exogenous Se treatment affected the changes in content of the endogenous phytohormones MeJA, GA7, and ETH in ginkgo leaves, promoted the expression of transcription factors such as *AP2*/*ERF*, *WRKY57,* and *WRKY41* in root tissues, and further affected the ginkgolides’ synthesis genes *GbDXRs*, *CYP450_ 724, CYP450_ 725,* and so on. The effect of exogenous Se on hormone synthesis in leaves was more obvious than that the effect that was observed in the roots. The hormone transport to the roots promoted the synthesis of ginkgolides and transported the ginkgolides back to the leaves.

## 4. Materials and Methods

### 4.1. Plant Materials and Growth Conditions

Three year-old ginkgo seedlings (Jiafoshou) were used in the experiment, and the seedling specifications were the same, with an average seedling height of 43.9 cm. The specification of the test basin was 25 cm × 17 cm (diameter × height), and it was a plastic basin. Each basin contaied about 3 kg of soil. The test soil was composed of river sand mixed with vermiculite and nutrient soil. The pH value was 5.5–7.0, the organic matter was ≥35.0%, and the contents of N, P, and K were 73.67 mg/kg, 55.41 mg/kg, and 62.37 mg/kg, respectively. The content of N, P, and K were determined in accordance with the description given in Gao et al. [43]. The sample seedling culture conditions were light for 14 h and they were dark for 10 h. The light intensity was 1000 lx and they were grown at 25 °C. One plant per pot was evenly planted, and 30 plants were treated in each sample.

Na_2_SeO_3_ was prepared into a 10, 15, 20, 25, 30 mg/L aqueous solution, and it was treated with foliar spraying (group A) and root irrigation (group B). The leaves were sprayed until they dripped, and the root treatments were irrigated with 500 mL per pot. Tween 80, at 0.01%, was added to the Na_2_SeO_3_ solution that was sprayed on the foliar, and the control group (group CK) was replaced with distilled water. After 12 weeks, the biomass, root to shoot ratio, and chlorophyll content were measured. In accordance with a previous study, 20 mg/L of a Na_2_SeO_3_ aqueous solution was selected for leaf spraying (group A) and root irrigation (group B); it would be used for the same treatment on the leaves and roots, respectively. Samples were taken every 2 weeks to determine the changes in soluble sugar, soluble protein and terpene lactone content.

In the 9th week after seedling treatment, 3 g leaves and 3 g taproots of three seedlings were randomly selected, and three biological replicates of each treatment were used to determine the content of different ginkgolides; transcriptome sequencing analysis was conducted. The samples were then stored in a refrigerator at −80 °C before they were sent for testing.

### 4.2. Physiological Index Measurement

The biomass increment was set as the biomass after the end of the test, minus the biomass before the treatment. The root to shoot ratio was the root biomass of the plant material divided by the aboveground biomass.

The top 2–3 functional leaves were selected to be measured with a FMS-2 portable modulated chlorophyll meter (Hansatech Co., Ltd., Pentney, UK), and the chlorophyll content was then converted.

The content of soluble sugar in plants was determined by anthrone colorimetry. The content of soluble protein was determined using the Coomassie Brilliant Blue G-250 method.

### 4.3. Determination of TTL Content in G. biloba Leaves

Ginkgo leaf samples treated with Na_2_SeO_3_ were dried for 48 h at 60 °C in a constant temperature drying oven. The dried ginkgo leaves were ground and packed separately for testing.

Ginkgo powder was accurately weighed at 1.5 g, and it was then put into a 250 mL round bottom flask. Fifty milliliters of 90 % methanol solution were added. Then, it was condensated and refluxed at 60 °C for 3 h. After the extraction, it was extracted and filtered while hot. It was then evaporated and concentrated on a rotary evaporator. A small amount of 50% methanol solution was then added to fully dissolve the concentrate. After multiple washings, the volume was fixed in a 10 mL volumetric flask. The test solution was subjected to 0.45 μM, and then the lactone content was detected by HPLC-ELSD. Ginkgolide A, ginkgolide B, ginkgolide C, ginkgolide J, and bilobalide (HPLC ≥ 98%) were used as single and mixed standard samples, and the peak time and standard curve were determined. The mobile phase was composed of methanol-tetrahydrofuran-water (20:8:72), and other conditions of chromatography were set in accordance with Zheng’s method [13]. The injection volume was 10 µL, and three replications were performed. The statistical significance of the TTLs content was analyzed with the GraphPad Prism 7 (San Diego, CA, USA). All tests were performed in triplicate, and data were represented as a mean ± SD (*n* = 3).

### 4.4. Transcriptome Sequencing and Analysis

The total RNA of the four stages was extracted using the Trizol Kit (Promega, Beijing, China), in accordance with the manufacturer’s instructions. The RNA was treated with DNase I (Takara to remove genomic DNA). The RNA integrity and quality were verified with RNase-free agarose gel and NanoDrop 2000 (IMPLEN, Westlake Village, CA, USA).

The total RNA was used as input material for the RNA sample preparations. The clustering of the index-coded samples was performed on a cBot Cluster Generation System using the TruSeq PE Cluster Kit v3-cBot-HS (Illumina, Metware Biotech Co., Ltd., Wuhan, China), in accordance with the manufacturer’s instructions. After cluster generation, the library preparations were sequenced on an Illumina Novaseq platform and 150 bp paired-end reads were generated. Raw data (raw reads) of a fastq format were first processed through in-house pearl scripts. Then, Q20, Q30, GC content, and the clean data were calculated. All the downstream analyses were based on the clean data, and they were of a high quality. The reference genome and gene model annotation files were directly downloaded from the genome website. The mapped reads of each sample were assembled using the StringTie (v1.3.3b) in a reference-based approach [44].

Transcriptome assembly was then performed using Trinity software. The gene function was annotated using the following databases: the National Center for Biotechnology Information’s non-redundant protein sequences (Nr), the Kyoto Encyclopedia of Genes and Genomes (KEGG), the Protein family (Pfam), the evolutionary genealogy of genes: Non-supervised Orthologous Groups (egg-NOG), Swiss-Prot, and Gene Ontology (GO). Fragments per kilobase of transcript per million fragments (FPKM) were used to represent the expression level of unigenes.

### 4.5. Determination of Phytohormone and Metabolome

The metabolome analysis of different samples of *G. biloba* was completed using Metware Biotech Co., Ltd., Wuhan, China (www.metware.cn, accessed on 5 January 2021) in accordance with Cheng’s method [45]. The freeze-dried leaves and roots were crushed using a mixer mill (mm 400, retsch) with a zirconia bead for 1.5 min at 30 Hz. One hundred milligrams of powder were weighted and extracted overnight at 4 °C, with 1.0 mL 70% aqueous methanol. Following centrifugation at 10,000× *g* for 10 min, the extracts were absorbed (CNWBOND Carbon-GCB SPE Cartridge, 250 mg, 3 mL; ANPEL, Shanghai, China) and filtrated (SCAA-104, 0.22 μm pore size; ANPEL, Shanghai, China) before LC-MS analysis.

The determination of phytohormones was based on the Metware Biotech Co., Ltd. LC-MS/MS platform. The API 6500 Q TRAP LC/MS/MS System, equipped with an ESI Turbo Ion-Spray interface, operated in a positive ion mode and was controlled using Analyst 1.6 software (AB Sciex, Toronto, ON, Canada). The ESI source operation parameters were as follows: ion source; turbo spray; source temperature: 500 °C; ion spray voltage (IS): 5500 V; curtain gas (CUR) was set at 35.0 psi; and the collision gas (CAD) was at a medium level. The DP and CE for individual MRM transitions were calculated using further DP and CE optimization. A specific set of MRM transitions were monitored for each period according to the phytohormones eluted within this period.

### 4.6. qRT-PCR Validation

The qRT-PCR was performed using cDNA templates. Each cDNA of the mRNA was amplified by qPCR using 2 × TSINGKE Master qPCR Mix (SYBR Green I) and a Goldenstar™ RT6 cDNA Synthesis Mix (TSINGKE, Xi’an, China).

Applied Biosystems (Foster City, CA, USA) using the SYBR-Green (Takara, Dalian, China) method were used. The primers were designed using the software of SnapGene v4.3.6 (San Diego, CA, USA) (Appendix A). The baseline and thresholdcycles (Ct) were automatically determined by the software of the system. Transcript levels were normalized against the average expression of the Actin gene.

### 4.7. Statistical Analysis

Excel 2021 and SPSS v.22.0 were used for experimental data processing. The heat map of expression analysis and hormone content analysis were processed using TB-tools software [46]. Duncan’s test was used to analyze significant differences (*p* ≤ 0.05). All of the experimental data had three biological replicates. The correlation network was performed using OmicStudio tools at https://www.omicstudio.cn/tool (accessed on 15 July 2022). The positive correlation threshold was set to be greater than or equal to 0.5, the negative correlation threshold was set to be less than or equal to −0.5, and the *p*-value threshold was less than 0.5 (R version 3.6.1, igraph1.2.6, Metware Biotech Co., Ltd., Wuhan, China) [45].

## 5. Conclusions

Ginkgolide is a unique terpenoid natural compound in *G. biloba*, which has important medicinal value. Se is not a necessary nutrient element for plant growth, but an appropriate amount of selenium can not only promote plant growth and development, but also improve plant quality, stress resistance, and disease resistance. At present, there are few reports concerning the effect of Se on the synthesis of plant terpenoids. In the present study, low-dose Na_2_SeO_3_ was found to promote the biomass of ginkgo leaves, increase the root to shoot ratio, and promote photosynthesis. At the same time, Se promoted the content of soluble sugar and soluble protein in the leaves; The synthesis of primary metabolites further promoted the synthesis of terpene lactones in *G. biloba* leaves. Transcriptome analysis showed that the DEGs caused by Se treatment were mainly enriched by phenylpropane biosynthesis, phytohormone signal transduction, amino acid and nucleotide sugar metabolism, and starch and sucrose metabolism. The results of the co-expression network analysis of TTLs and DEGs showed that the synthesis of ginkgolide A, ginkgolide B, ginkgolide M, and bilobalide was affected by the expression levels of *GbDXR*, *GbMECPs*, *GbWRKY*, *GbAP2*/*ERF*, and so on. The phytohormones MeJA-ILE, ETH, and GA7 also participated in this process and were found to be significantly positively correlated with gene expression. Considering that terpene lactones are synthesized in the roots of *G. biloba* and endogenous hormones are synthesized in the leaves, the low dose Se treatment of ginkgo leaves caused changes in the levels of MeJA-ILE, ETH, GA7 and other hormones, which, in turn, promoted the expression of TFs such as *WRKY* and *AP2*/*ERF*, thereby inducing the expression of structural genes, which thus promoted the synthesis of root terpene lactones, and transported them to *G. biloba* leaves for storage.

## Figures and Tables

**Figure 1 molecules-27-07548-f001:**
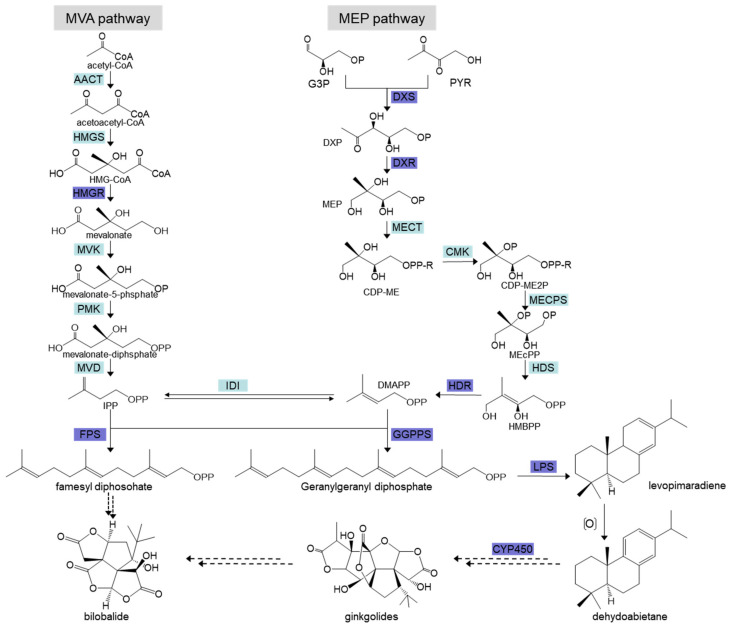
Biosynthetic pathways of ginkgolides and bilobalide in *Ginkgo biloba*.

**Figure 2 molecules-27-07548-f002:**
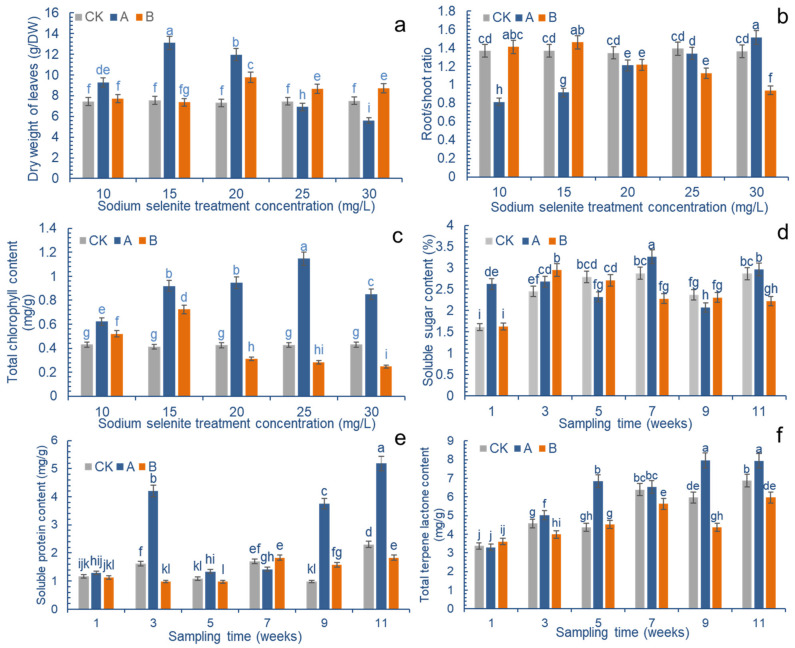
Effects of Na_2_SeO_3_ on the physiological indexes and total terpene lactone content in Ginkgo leaves. Na_2_SeO_3_ was prepared in 10, 15, 20, 25, 30 mg/L aqueous solutions, and treated with foliar spraying (group A) and root irrigation (group B). Tween 80, at 0.01%, was added to the sodium selenite solution that was sprayed on the foliar, and the control group (group CK) was replaced with distilled water. Regarding the Na_2_SeO_3_ solution, 20 mg/L was used for foliar spraying (group A) and root irrigation (group B), for the same treatment, and samples were taken every two weeks to determine the changes in soluble sugar, soluble protein, and terpene lactone content. The lowercase letters a–l on the bar reflects the 5% significant level. (**a**) Total biomass of the treated ginkgo leaves; (**b**) the root to shoot ratio; (**c**) change in total chlorophyll content in leaves; (**d**) changes in soluble sugar content; (**e**) changes in soluble protein content; and (**f**) change in total terpene lactone content.

**Figure 3 molecules-27-07548-f003:**
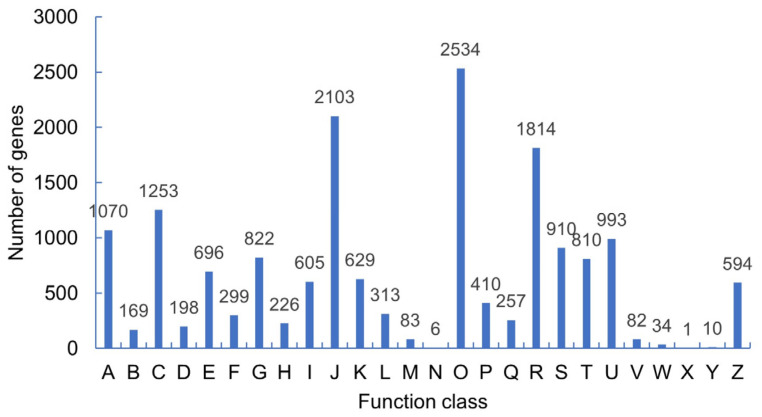
KOG classification of the DEGs. (A) Processing and modifying RNA; (B) chromosome structure and changes; (C) energy generation and transformation; (D) cell cycle regulation, division, and chromosome rearrangement; (E) amino acid transport and metabolism; (F) nucleotide transport and metabolism; (G) carbohydrate transport and metabolism; (H) coenzyme transport and metabolism; (I) lipid transport and metabolism; (J) translation, ribosome structure, and biosynthesis; (K) transcription; (L) replication, reorganization, and repair; (M) cell wall and membrane biosynthesis; (N) cell movement; (O) post translational modification, protein foldings and molecular chaperone; (P) inorganic ion transport and metabolism; (Q) synthesis, transportation, and metabolism of secondary metabolites; (R) general function prediction; (S) unknown function; (T) signal transduction mechanism; (U) intracellular transport, secretion, and vesicular transport; (V) defense mechanism; (W) extracellular movement; (X) unnamed protein; (Y) nucleic acid structure; and (Z) cytoskeleton.

**Figure 4 molecules-27-07548-f004:**
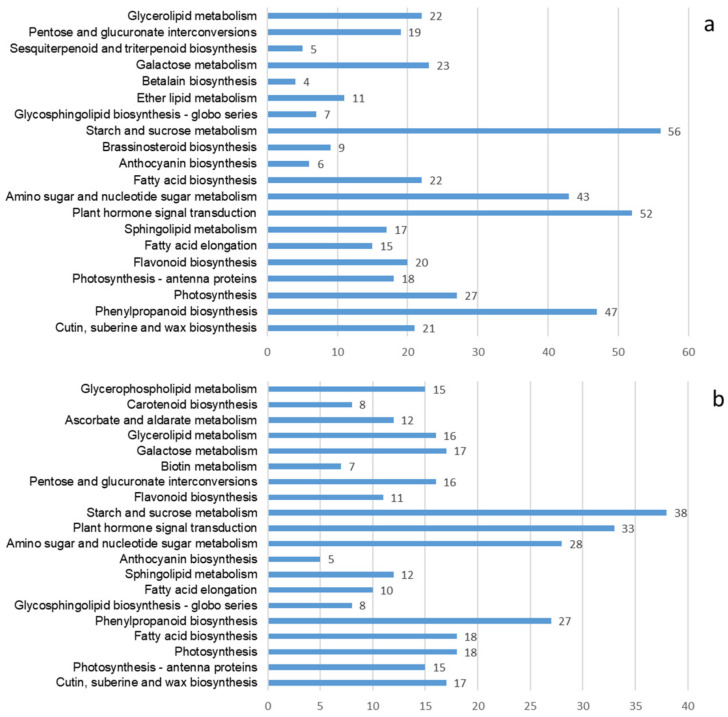
Annotation analysis of the KEGG pathway of differentially expressed genes in ginkgo leaves treated with Na_2_SeO_3_. (**a**) Group A vs. CK; (**b**) group B vs. CK. Note: CK control group; A—foliar spraying; B—root pouring.

**Figure 5 molecules-27-07548-f005:**
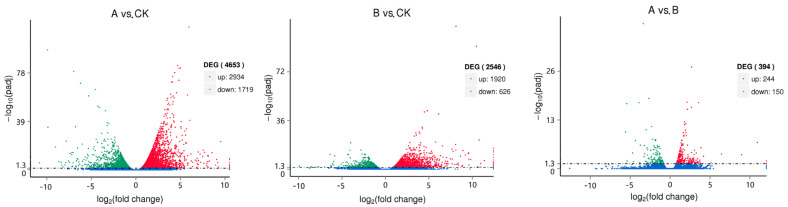
Volcanic map of differentially expressed genes in foliar and root selenium application treatments. Note: CK control group; A—foliar spraying; B—root pouring.

**Figure 6 molecules-27-07548-f006:**
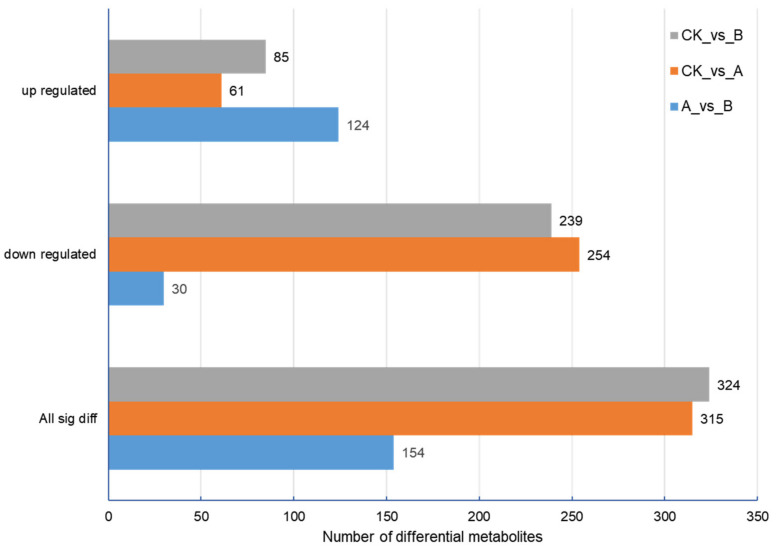
Statistics concerning the number of differential metabolites.

**Figure 7 molecules-27-07548-f007:**
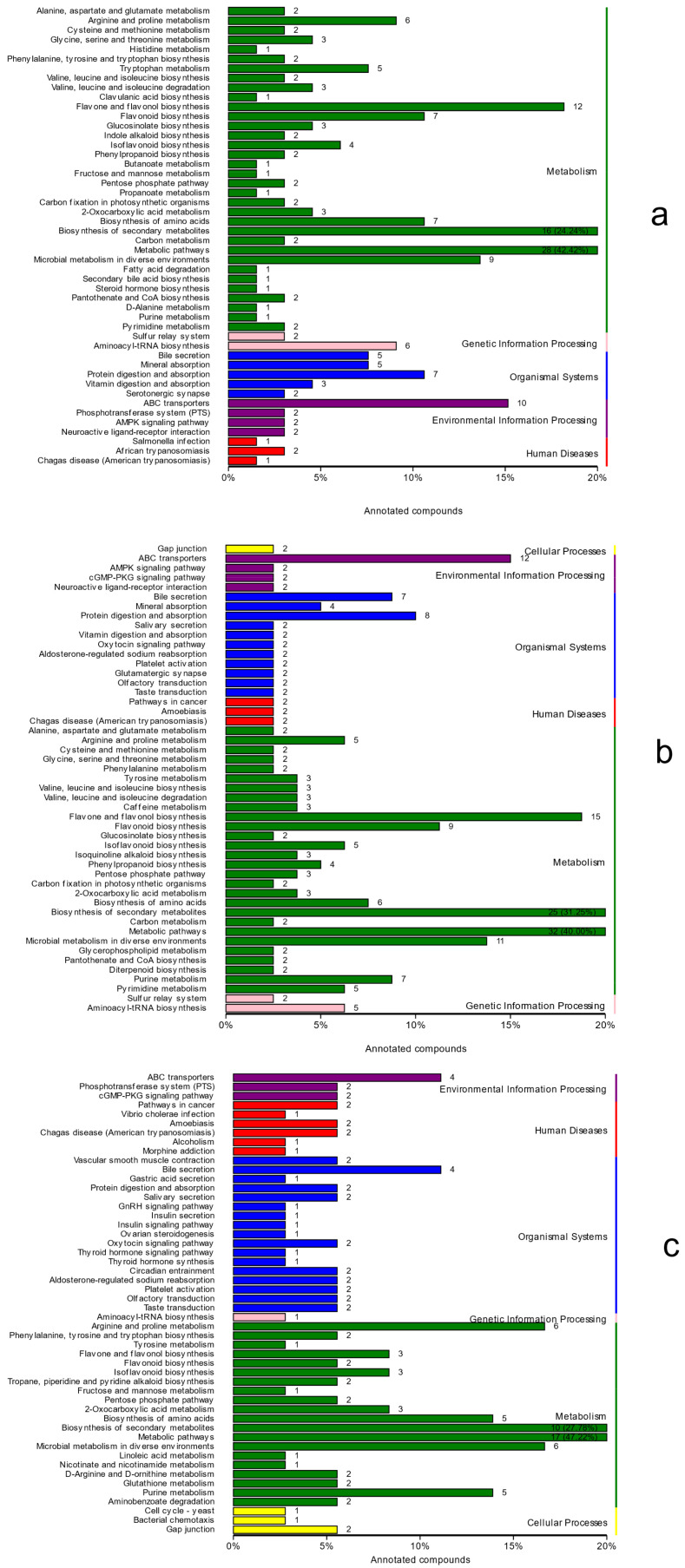
Classification diagram of KEGG, a differential metabolite. (**a**) CK vs. A; (**b**) CK vs. B; (**c**) A vs. B. Note: The ordinate is the name of the KEGG metabolic pathway, and the abscissa refers to the number of metabolites that were annotated on the pathway as a proportion of the total number of metabolites that were annotated.

**Figure 8 molecules-27-07548-f008:**
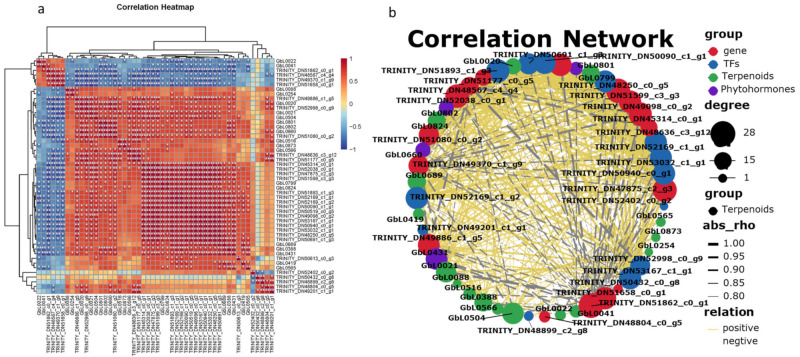
Correlation analysis between differentially expressed genes and metabolites that are related to terpene synthesis in ginkgo leaves. (**a**) Correlation clustering heat map between core differential metabolites and differentially expressed genes; (**b**) correlation network diagram between ginkgolides, differential hormones, and hot genes. ** represents very significant level; * represents significant level.

**Figure 9 molecules-27-07548-f009:**
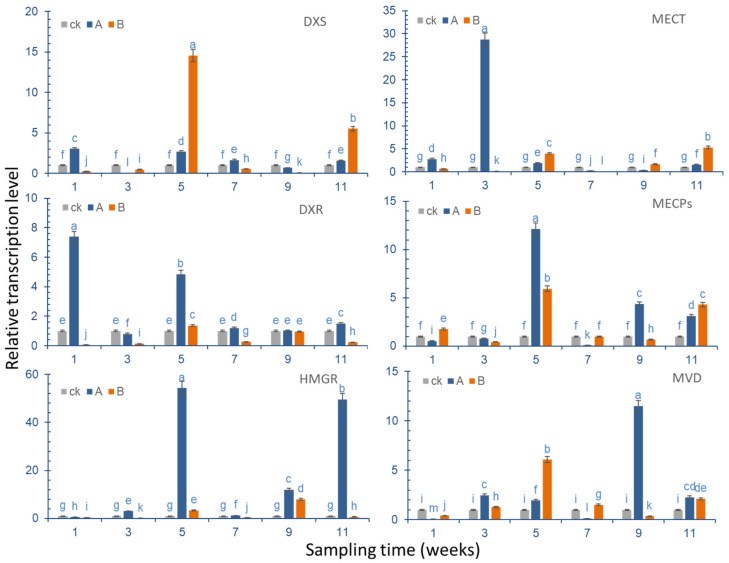
Relative expression level of six differentially expressed genes in the ginkgo terpene lactone synthesis pathway at different sampling points. Note: CK. control group; A. foliar spraying; B. root pouring. The lowercase letters a–m on the bar reflects the 5% significant level.

## Data Availability

This transcriptome and metabolome data of *Ginkgo biloba* has been deposited in China National Center for Bioinformation (CNCB)—file ID: OMIX002229.

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
