# Peer review of "Treatment of Ginkgo biloba with Exogenous Sodium Selenite Affects Its Physiological Growth, Changes Its Phytohormones, and Synthesizes Its Terpene Lactones"

_molecules, 2022, doi:10.3390/molecules27217548_

Round 1
Reviewer 1 Report
Line no. 17- replace root shoot ratio with root to shoot ratio
Arrange the keyword in alphabet order
Introduction is fine. I suggest add some more information regarding gene expression and regulation.
Results- good presentation
Discussion- appropriate
Line no. -334. Se can reduce the biomass of leaves; use full stop.
Line 342-use root to shoot ratio
Line no.-384 Bilobalide-------B should be small letter
Material and methods-
I would like to say that please mention ginkgo seedlings cultivars name (wild or cultivated , if cultivated then mention variety/genotypes name .
Samples were taken every 2 weeks to determine the changes of soluble sugar, soluble pro- 441 tein and terpene lactone content.- Please mention full detail of methodology of each components. What about samples replication? how you collected samples? did you mix samples of each treatments or collected replication wise of each treatments for analysis. It is not clear here.
References
Please match references with text.
Overall research work is interesting for public domain. It can be considered for publication after inclusion of suggested comments.
Author Response
Dear Reviewer,
Thank you for your advice and help. I have revised the paper as follows,
- We have replace the root shoot ratio to “root to shoot ratio”in the full paper;
- Adjusted keyword order alphabetically;
- In the introduction, we have added some report onthe regulation and expression of key genes for ginkgolide synthesis;
- We have added the ginkgo cultivars name in the material section;
- The repetition of experimental sampling and the amount of reagent to be treated are also explained and supplemented in the materials;
- We also polished the language of the whole article;
- Other modification details have been marked in blue for your reference.
Thank you again!
Reviewer 2 Report
1. The English language needs to be strongly improved by a professional editing service.
2. The sequence of A B and CK should be unified in Figure 1. CK on the left, A and B on the right, that is, a b c should be consistent with d e f in Figure 1.
3. Lack of significance analysis in bar chart of Figure 1and 8.
4. It is suggested to include a mechanistic map or simple metabolic pathway (with indicated enzyme genes).
5. Among the methods, the determination of N P K and organic matter is not mentioned. Please supplement metabolome determination and analysis methods.
6. The concentration of sodium selenite treatment is indicated in the method, but the dosage is not indicated. Please add.
7. Please upload transcriptome and metabolome data to NCBI or other database.
Author Response
Thank you for your advice and help. I have revised my paper as follows,
- We have polished the language of the whole articlein a translation company;
- The order of the histogram in Figure 1 has been adjusted, and the difference significance analysis has been added in Figure 1 and Figure 8;
- In the introduction section, we added the reference metabolic diagram of ginkgolide biosynthesis as Figure 1;
- In the material part, we added specific parameters of nitrogen, phosphorus and potassium content in plant culture media;The dosage used for sodium selenite treatment of samples was also supplemented;
- In the Data Availability Statement, we have upload the transcriptome and metabolome data to CNCB database, and the file ID was OMIX002229;
- All changes in the manuscriptare marked in blue word, please refer to.
Thank you again!
Round 2
Reviewer 2 Report
1. The sequence of A B and CK in Figure 9 should be consistent with Figure 2. That is, in Figure 9 the sequence should be CK A (in bule bar) and B (in red bar).
2. Please performed significance analysis between different concentration treatments in Figure 2 a b and c respectively, not just in one treatment. Please performed significance analysis between different weeks treatments in Figure 2 d e and f respectively, not just in one treatment. Same in Figure 9.
3. Use Figures of this manuscript in the discussion.
4. Among the methods, the determination of N P K and organic matter is not mentioned. Please supplement metabolome determination and analysis methods, or references.
5. In Figure 9, the DXR relative transcription level of B significantly higher than other treatments, however, the total terpene lactone of B significantly lower than other treatments. As DXR plays an important role in the biosynthesis of ginkgolide, please explain why.
Author Response
Dear Editor and Referees, Thank you for your valuable comments, which are very helpful to the promotion of manuscripts. We made the following modifications according to your suggested, and used the track changes in MS Word, 1. We adjusted the processing order of Figure 1 and Figure 9 to be consistent, and added the overall significance analysis respectively. 2. A description of the result picture was added to the discussion; 3. In the part of materials and methods, we added the references of soil N, P and K determination; In addition, the reference method for the determination of metabolome and phytohormone and the testing company are added; 4. By analyzing the original data, we found that there was an error in the order of root and leaf processing record data in the mapping of qRT-PCR of DXR gene, which has been corrected in the images and results. Best! Hua